# The Eukaryotic Translation Initiation Factor 4F Complex Restricts Rotavirus Infection via Regulating the Expression of IRF1 and IRF7

**DOI:** 10.3390/ijms20071580

**Published:** 2019-03-29

**Authors:** Sunrui Chen, Cui Feng, Yan Fang, Xinying Zhou, Lei Xu, Wenshi Wang, Xiangdong Kong, Maikel P. Peppelenbosch, Qiuwei Pan, Yuebang Yin

**Affiliations:** 1Biomedical Research Center, Northwest Minzu University, Lanzhou 730030, China; s.chen.1@erasmusmc.nl (S.C.); q.pan@erasmusmc.nl (Q.P.); 2Erasmus MC-University Medical Center, 3000 CA Rotterdam, The Netherlands; w.wang.2@erasmusmc.nl (W.W.); m.peppelenbosch@erasmusmc.nl (M.P.P.); 3Department of Materials Engineering, Zhejiang Sci-Tech University, Hangzhou 310018, China; fengc93103@126.com (C.F.); kxd01@126.com (X.K.); 4College of Basic Medicine, Shannxi University of Chinese Medicine, Xianyang 712046, China; fangyan9494@163.com; 5Institute of Molecular Immunology, School of Laboratory Medicine and Biotechnology, Southern Medical University, Guangzhou 510515, China; xylona14@hotmail.com; 6College of Life Sciences, Northwest A&F University, Yangling 712100, China; xulei@nwafu.edu.cn

**Keywords:** rotavirus, eIF4F complex, translation initiation factors, IRF1, IRF7

## Abstract

The eIF4F complex is a translation initiation factor that closely regulates translation in response to a multitude of environmental conditions including viral infection. How translation initiation factors regulate rotavirus infection remains poorly understood. In this study, the knockdown of the components of the eIF4F complex using shRNA and CRISPR/Cas9 were performed, respectively. We have demonstrated that loss-of-function of the three components of eIF4F, including eIF4A, eIF4E and eIF4G, remarkably promotes the levels of rotavirus genomic RNA and viral protein VP4. Consistently, knockdown of the negative regulator of eIF4F and programmed cell death protein 4 (PDCD4) inhibits the expression of viral mRNA and the VP4 protein. Mechanically, we confirmed that the silence of the eIF4F complex suppressed the protein level of IRF1 and IRF7 that exert potent antiviral effects against rotavirus infection. Thus, these results demonstrate that the eIF4F complex is an essential host factor restricting rotavirus replication, revealing new targets for the development of new antiviral strategies against rotavirus infection.

## 1. Introduction

Rotavirus is considered to be one of leading causative agent of severe diarrhea in infants younger than five years old [1], and it causes estimated 215,000 deaths in children each year globally [2]. Although rotavirus infection mainly occurs in low-income countries [3], it also inflicts a heavy burden in industrialized countries. For example, in the European Union, rotavirus infection causes more than 200 deaths, over 87,000 hospital admissions, and almost 700,000 outpatient visits in children younger than five years of age annually [4]. Emerging evidence indicates that rotavirus infection causes severe complications in organ transplant patients irrespective to their ages [5]. Although vaccines have been developed, no approved antiviral treatment is available.

The genome of rotavirus contains 11 segments encoding 12 proteins including six structural (VP1-4, VP6, and VP7) and six non-structural proteins (NSP1-6) [6]. Among the structural proteins, as a spike protein, rotavirus VP4 plays an essential role in both viral entry and exit [7]. VP4 was also demonstrated to be of importance in viral attachment and internalization [8], which is often used for the development of rotavirus vaccines [8]. VP4 contains two subunits including a C-terminal subunit VP5* and a N-terminal subunit VP8*, and both VP5* and VP8* help virus entry by interacting with several putative partners and cell surface receptors [7]. The rotavirus genome is a double-strand RNA containing a cap at 5′ untranslated regions (UTR) synthesized by the viral transcriptase but lacks a polyadenylated tail instead having a consensus sequence at 3′ UTR [9]. It has been reported that rotavirus NSP3 is able to bind to the 3′ consensus sequence of viral mRNA and interact with eIF4G to aid translation of viral mRNA [10]. Rotavirus NSP3 stabilizes the eIF4E–eIF4G interaction to exert an enhanced effect on the translation of both poly (A)- and non-poly (A)-tailed mRNAs [11].

There are three phases in protein synthesis including initiation, elongation, and termination [11]. Initiation determines translation rates [12]. Most of the eukaryotic mRNAs are characterized by a m7GpppX structure (where m7Gppp is the 7-methyl-guanosine-containing and X is any nucleotide), termed as a cap, at the 5′ ends and the poly (A) tail at the 3′ end [13]. The mRNA containing a cap at the 5′ ends is able to be more efficiently translated than that lacking this structure [14], and these mRNAs are translated in a cap-dependent manner [12]. The cap structure is bound by the eukaryotic translation initiation factor 4F (eIF4F) which is a protein complex containing three constituent proteins: a eukaryotic translation initiation factor 4A (eIF4A), a eukaryotic translation initiation factor 4E (eIF4E), and a eukaryotic translation initiation factor 4G (eIF4G) [15]. The eIF4F complex plays a pivotal role in cap-dependent mRNA protein translation initiated by recruiting mRNA to a ribosome [13]. As an RNA helicase, eIF4A makes use of ATP hydrolysis to unwind and resolve the RNA secondary structure [16]. The eIF4E recognizes and binds to the 7-methylguanosine (m7G) cap located at the 5′-UTRs of mRNA to mediate the mRNA recruitment on ribosomes, thus initiating the translation together with other initiation factors [17]. As a scaffold protein, eIF4G plays a central role in translation initiation by assembling eIF4E and eIF4A to further form the eIF4F complex [18]. The programmed cell death protein 4 (PDCD4) is a translation suppressor of mRNAs by interacting with eIF4A to suppress its helicase activity [19]. As a downstream target of mechanistic target of the rapamycin (mTOR) pathway, the phosphorylation of protein kinase S6K1 is able to phosphorylate PDCD4 to release it from eIF4A, thus allowing eIF4A to interact with eIF4G to form the eIF4F complex, followed by initiating translation [20].

Viruses require components from the host cell to replicate, assemble viral components, and release their new synthesized virions [14]. Viral protein synthesis completely relies on the translational machinery of the host due to viruses lacking this machinery themselves [21]. Viruses are able to exploit the translational machinery, including eIF4F, to support the translation of their transcripts [15]. Certain mammalian viruses are involved in targeting the eIF4F complex to regulate both viral and host mRNA translation; while many RNA viruses inhibit host protein synthesis to initiate the translation of their own mRNAs by inhibiting eIF4F [22]. Feline calicivirus (FCV) and mouse norovirus (MNV) have been reported to be capable of directly interacting with eIF4E, and viral RNA translation requires the eIF4A, indicating that the replication of the two viruses requires eIF4F [23]. Blocking eIF4E–eIF4G interaction has been demonstrated to cause inhibition of coronavirus replication, indicating that eIF4F has a promoting effect on the virus replication [24]. Furthermore, certain members of the eIF4G family were thought to be able restrict the infection of Rice yellow mottle virus (RYMV) [25]. Thus, the eIF4F complex has different effects on distinct viruses.

The eIF4F complex plays an essential role in regulating interferon signaling which is considered to be the first line of antiviral defense [26]. The phosphorylation of eIF4E was reported to exert antiviral effect via regulating the production of type I interferon [27]. It was found that the eIF4F complex could tightly regulate the translation of the signal transducer and activator of transcription 1 (STAT1) mRNA that plays a crucial role in type I interferon signaling [28]. Thus, in this study, we have dissected the effects of the cellular translation machinery, the eIF4F complex, on rotavirus infection, and we found that the eIF4F complex is an essential host factor in counteracting rotavirus infection through regulating the antiviral protein IRF1 and IRF7.

## 2. Results

### 2.1. The eIF4A Inhibits Rotavirus Infection

We first examined the effects of rotavirus infection on the expression of the components of the eIF4F complex, including eIF4A and eIF4E, at indicated time points (1, 2, 4, 6, 24, and 48 h). Although rotavirus infection did not affect the protein expression of the eIF4F complex (Appendix A), it is interesting to investigate whether the components of the eIF4F complex regulate the course of the rotavirus infection. To this aim, we first detected the effects of eIF4A on rotavirus infection, by performing a lentiviral RNAi-mediated loss-of-function assay to silence the eIF4A gene. All five short hairpin RNA (shRNA) vectors showed successful knockdown (Figure 1A,B). Importantly, two shRNA vectors (No 1 and 2) resulted in a 121.7 ± 60.6 (*n* = 6, *P* < 0.01)- and 56.7 ± 29.9 (*n* = 6, *P* < 0.01)-fold increase in SA11 rotavirus mRNA, respectively (Figure 1C). All shRNA vectors showed no clear cytotoxicity determined by 3-(4,5-dimethyl-2-thiazolyl)-2,5-diphenyl-2-H-tetrazolium bromide (MTT) assay (Appendix A). Western blot indicated that all five shRNA vectors increased the rotavirus VP4 protein level (Figure 1D). To further verify these findings, we used CRISPR/Cas9 to knockout the eIF4A gene in Caco2 cells (Figure 1E). After eIF4A knockout, the viral mRNA and protein levels were remarkably reduced (Figure 1F,G).

### 2.2. Blocking eIF4E Promotes Rotavirus Infection

Another component of the eIF4F complex, eIF4E, is responsible for binding to the cap of mRNA. Similarly, a lentiviral RNAi-mediated loss-of-function assay was performed to silence the eIF4E gene to assess its effect on rotavirus infection. One out of the four shRNAs exerted a potent knockdown effect on eIF4E (Figure 2A). Consistently, this shRNA vector resulted in a 14.5 ± 5.8 (*n* = 6, *P* < 0.01)-fold increase in SA11 rotavirus mRNA (Figure 2B). All shRNA vectors showed no clear cytotoxicity measured by MTT assay (Appendix A). This shRNA vector resulted in the increase in the SA11 rotavirus VP4 protein level (Figure 2C). To further verify this, we used CRISPR/Cas9 to knockout eIF4E in Caco2 cells (Figure 2D). After eIF4E gene knockout, the viral RNA and protein levels were restricted (Figure 2E,F).

### 2.3. The eIF4G Suppresses Rotavirus Infection

As a scaffolding protein, eIF4G protein is able to physically link mRNA and the small ribosomal subunit via protein-protein interactions. To investigate the effects of eIF4G protein on rotavirus infection, a lentiviral shRNA-mediated loss-of-function assay was conducted, which resulted in gene silence of eIF4G by all five shRNA vectors at both the mRNA and protein level (Figure 3A,B). Importantly, all these shRNA vectors resulted in a potent increase in the viral RNA and protein level (Figure 3C,D). The vitality of cells was not affected by these shRNA vectors (Appendix A). To further verify these findings, we performed a Crispr/cas9 assay to knockout eIF4G in Caco2 cells (Figure 3E). After gene knockout, the viral RNA and protein levels were inhibited (Figure 3F,G).

To further verify the effect of eIF4A, eIF4E, and eIF4G on rotavirus infection, CRIPRs against eIF4A, eIF4E, and eIF4G were co-transfected in Caco2 cells. After puromycin selection, the knockout of eIF4A, eIF4E, and eIF4G was detected by western blot, indicating successful knockout of three genes (Figure 4A). The effect of eIF4A, eIF4E, and eIF4G on rotavirus infection was further detected by RT-qPCR and western blot, demonstrating that both viral genomic RNA and viral protein VP4 synthesis were promoted (Figure 4B,C).

### 2.4. Programmed Cell Death Protein 4 (PDCD4) Promotes Rotavirus Infection

PDCD4 functions by inhibiting the translation by directly inhibiting the helicase activity of eIF4A to competitively bind to the scaffold protein eIF4G. To further verify the effects of the eIF4F complex on rotavirus infection, a lentiviral shRNA-mediated loss-of-function assay was performed to silence PDCD4, which indicated that one out of three shRNA vectors (No 1) exerted a potent knockdown effect (Figure 5A,B). Consistently, the shRNA vector resulted in a 50 ± 12 % (n= 8, *P* < 0.05)-fold reduction in SA11 rotavirus mRNA (Figure 5C), and remarkable reduction in viral VP4 protein levels (Figure 5D). All shRNA vectors have no significant cytotoxicity (Appendix A).

### 2.5. The eIF4F Complex Inhibits the Level of Antiviral Proteins to Exert Anti-Rotavirus Effect

To further explore the underlying molecular mechanism of the antiviral effects of the eIF4F complex on rotavirus infection, we questioned the effect of the eIF4F complex on several antiviral proteins including interferon regulatory factor 1 (IRF1), IRF7, and retinoic acid-inducible-I (RIG-I), which indicated that the knockdown of eIF4A had no obvious effect on these antiviral proteins. While the knockdown of eIF4E (shRNA No 2) was able to inhibit protein level of IRF1 and IRF7 (Figure 6B). Knockdown of eIF4G was also capable of suppressing the protein level of IRF1 and IRF7 (Figure 6C). To further verify this, the effect of IRF7 and IRF1 on rotavirus replication was detected, indicating that knockout of either IRF7 (Figure 6D) or IRF1 (Figure 6F) could significantly reduce the genomic mRNA of SA11 rotavirus (Figure 6E,G). Thus, the antiviral effect against rotavirus of the eIF4F complex may operate through regulating the expression antiviral proteins in the host.

## 3. Discussion

Many viral mRNAs have acquired a variety of sophisticated strategies to compete with cellular mRNAs that are already present in the cytoplasm. This allows the selective translation of viral mRNAs, since they do not possess the required components to initiate the translation of mRNA [23,29]. The eIF4F initiation factor complex plays a vital role in initiating translation by recognizing the 5′ cap structure in a cap-dependent manner [23]. As the target of numerous cellular signaling pathways, eIF4F closely controls the translation in response to a multitude of environmental conditions including viral infection [30]. In contrast, viral infections often cause dysregulation of eIF4F expression to benefit its replication. Vesicular stomatitis virus (VSV) was reported to cause modifications of the eIF4F complex to inhibit host protein synthesis [31]. Human Cytomegalovirus (HCMV) replication was demonstrated to increase the overall abundance of the eIF4F components and promote assembly of the eIF4F complexes [32]. The cellular cap-dependent protein level was reported to be rapidly inhibited by foot-and-mouth disease virus (FMDV) via inducing the cleavage of eIF4G [33]. However, we did not find remarkably altered expression levels of the components of eIF4F, including eIF4A and eIF4E, upon rotavirus infection (Appendix A).

The eIF4F is a tripartite complex composed of three components including a cap-binding subunit (eIF4E), an RNA helicase (eIF4A), and a large molecular scaffold (eIF4G) that is associated with eIF3-bound 40S ribosome subunits [30]. These three components of the eIF4F complex were reported to closely regulate the life cycle of various viruses either promoting or restricting the infection. By treating influenza virus infected cells with Hippuristanol, a sterol, which is able to suppress RNA binding, ATPase, and helicase activities to block eIF4A-dependent translation, which demonstrated that eIF4A fully determines the translation of influenza virus mRNAs [34]; however, eIF4E does not affect influenza virus protein synthesis [29]. Hepatitis E virus (HEV) requires all three components of eIF4F to replicate in the host [35]. The protein synthesis of Junin virus (JUNV), Tacaribe virus (TCRV), and Pichinde virus (PICV) requires the participation of eIF4A and eIF4G but not eIF4E [36]. The eIF4F complex is required for maintaining translation efficiency of HCMV at the start of infection; while viral protein synthesis becomes increasingly insensitive to inhibition of eIF4F complex at late stage of virus infection [37]. Cencic et al. demonstrated that blocking the interaction between eIF4E and eIF4G by a small molecule was capable of remarkably suppressing corovarius replication [24]. Rotavirus mRNAs are capped but not polyadenylated, and its NSP3 is capable of binding to eIF4G to stabilize the eIF4E–eIF4G interaction [10]. However, knockdown of NSP3 did not influence the synthesis of viral proteins, indicating that it is not involved in viral mRNA translation and virus replication [38]. We have previously confirmed that the PI3K-Akt-mTOR pathway sustains rotavirus infection via its downstream effector 4E-BP1 [39], which provokes us to further investigate the effect of eIF4E on rotavirus infection, since mTOR exerts the function of splitting the 4E-BP1-eIF4E dimer via phosphorylating 4E-BP1 [40]. Unexpectedly, we found that blocking the three components of the eIF4F complex promotes rotavirus infection (Figure 1, Figure 2, Figure 3, and Figure 4). This suggests that the inhibitory effect of eIF4F on rotavirus might be via another scenario other than the regulating of the capping process of viral mRNA. PDCD4 is capable of directly inhibiting the helicase activity of eIF4A by competing its binding to eIF4G to suppress the activity of the eIF4F complex [41]. Consistently, we found that blocking PDCD4 was capable of inhibiting rotavirus infection (Figure 5), which is in agreement with the effect of the eIF4F complex on rotavirus infection. Herein, our findings suggest that the eIF4F complex exerts inhibitory effects on rotavirus infection.

Mounting evidence reveals that the eIF4F complex perverts the interferon signaling pathway that exerts a pivotal role in defensing various viruses. Viruses often trigger a cascade of events to activate certain transcription factors, such as IRF3, IRF7, and NF-κB, after being recognized by pattern-recognition receptors (e.g., Toll-like receptors), which eventually enhances interferon-β (IFN-β) to exert antiviral effects [27]. It was confirmed that the dephosphorylation of eIF4E was able to stimulate the NF-κB pathway to promote the production of type I interferon [27]. The eIF4F complex was demonstrated to closely regulate the translation of STAT1 mRNA [28]. In 4E-BP1^−/−^ and 4E-BP2^−/−^ double knockout mouse embryonic fibroblasts (MEF), it was found that type-I IFN was strongly promoted through enhancing the expression of IRF7 mRNA [26]. The binding of eIF4E to eIF4G may be involved in the translation of IRF7 mRNA [42]. Consistently, we demonstrated that knockdown and knockout of eIF4E and eIF4G were capable of suppressing the protein level of IRF1 and IRF7 (Figure 6). Importantly, we confirmed that IRF1 and IRF7 *per se* were able to significantly inhibit rotavirus replication (Figure 6). Thus, we have provided a possible novel insight into how the eIF4F complex regulates rotavirus replication.

In summary, this study has demonstrated that the eIF4F complex can remarkably inhibit rotavirus infection. The negative regulator of the eIF4F complex, PDCD4, increases rotavirus replication. The effect of the eIF4F complex on rotavirus replication operates by regulating the expression of certain antiviral proteins, including IRF1 and IRF7, which exert remarkable antiviral effect against rotavirus. Hereto, this study revealed a new insight into the regulation of eIF4F on rotavirus replication, and the eIF4F complex may represent as a promising target for the development of new antivirals and antiviral therapies against rotavirus infection.

## 4. Materials and Methods

### 4.1. Cell Culture and Virus

All cells used in the study were grown in Dulbecco’s Modified Eagle Medium (DMEM, Lonza, Verviers, Belguim) containing 20% (*v*/*v*) heat-inactivated fetal calve serum (FCS) (Hyclone, Lonan, Utah) and penicillin (100 IU/mL)/streptomycin (100 mg/mL) (P/S) Invitrogen-Gibco) at 37 °C in a humidified 5% CO_2_ incubator.

Simian rotavirus SA11 was used and prepared as described previously [39,43].

### 4.2. Inoculation of SA11 Rotavirus in Cells

The inoculation method of SA11 rotavirus was described previously [6,44]. Briefly, cell monolayers were incubated with SA11 rotavirus (MOI = 0.7) at 37 °C with 5% CO_2_ for 60 min for infection, followed by removing unbound viral particles by washing with cold PBS three times, followed by adding culture medium (FCS free) containing 5 μg/mL of trypsin and indicative drugs. Thereafter, cells were incubated at 37 °C in a humidified 5% CO_2_ incubator. Viral titer was detected by RT-qPCR after 48 h inoculation.

### 4.3. Real-Time Quantitative Reverse Transcription PCR (RT-qPCR) Analyses

Total RNA was isolated using NucleoSpin^®®^ RNA kit (MACHEREY-NAGEL, Düren, Germany) according to the manufacturer’s instructions. The extracted RNA was dissolved in diethylpyrocarbonate (DEPC)-treated water, and its concentration was measured using a Nanodrop ND-1000 (Wilmington, DE, USA). The 500 ng of total RNA was used for cDNA synthesis using the reverse transcription system from TAKARA (TAKARA BIO INC) according to manufacturer’s instructions. The resultant cDNA was diluted and used for evaluating gene expression with corresponding primers. All RT-qPCR experiments were performed by SYBR-Green-based (Applied Biosystems SYBR Green PCR Master Mix; Thermo Fisher Scientific Life Scienc; Carlsbad, CA, USA) real-time PCR with the StepOnePlus System (Thermo Fisher Scientific Life Sciences). The expression of target mRNAs was normalized to the glyceraldehyde 3-phosphate dehydrogenase (GAPDH) mRNA. The relative mRNA levels of target genes were expressed as 2^−ΔΔC^_t_, in which ΔΔC_t_ = (ΔC_tTarget_ − ΔC_tGAPDH_)_treatment_ − (ΔC_tTarget_ − ΔC_tGAPDH_)_control_. Gene primers for RT-qPCR were listed in Appendix A.

### 4.4. Gene Knockdown by shRNA

Lentiviral shRNA vectors, targeting eIF4A, eIF4E, eIF4B, and PDCD4 or non-targeted control lentivirus was prepared in 293T cells in a 6-well-plate using a standard third-generation packaging system (All shRNA primers were listed in Appendix A).

Cell monolayers were transfected with lentiviral vectors for three days, followed by screening by puromycin (8 μg/mL) to obtain stable clones, and the knockdown effect was detected by western blot.

### 4.5. Gene Knockout by Clustered Regularly Interspaced Short Palindromic Repeats (CRISPR/Cas9) Assay

The sgRNAs against eIF4A, eIF4E, eIF4G (primers were listed in Appendix A) were designed using freely available online tools (https://www.genscript.com/gRNA-database.html). Oligos were annealed according to the manufacturer’s protocol (37 °C for 30 min; 95 °C for 5 min and then ramp down to 25 °C at 5 °C/min). Subsequently, annealed oligos were ligated into Lenti CRISPR v2, followed by transforming 10 µL of electrocomp™ cells (ThermoFisher SCIENTIFIC, Carlsbad, USA) with 5 µL of annealed oligos, and then incubated at 37 °C for overnight. 2–3 colonies were picked up and cultured in LB medium, followed by performing a sequence verification. The verified colony was expanded in a LB containing Ampicillin culture (100 μg/mL, SIGMA-ALDRICH, Saint Louis, USA) and DNA was isolated by using DNA extraction kit (ThermoFisher SCIENTIFIC) according to manufacturer’s protocol. CRISPR DNA was co-transfected with packaging plasmids including pVSV-G, pMD, and pREV into HEK293T cells and lentiviral vectors were harvested.

Cell monolayers were transfected with lentiviral vectors for three days, followed by screening by puromycin (8 μg/mL) to obtain stable clones, and the knockdown effect was detected by western blot.

### 4.6. The 3-(4, 5)-Dimethylthiahiazo (-z-y1)-3, 5-di- Phenytetrazoliumromide (MTT) Assay

MTT assay was used to evaluate cell viability, and the cells were seeded into 96-well plates (5000 cells/well). Then, the viable cells were quantified at indicated time points by adding 10 μL 5 mg/mL MTT per well, and the cells were incubated for an additional 3 h in a humidified incubator. Thereafter, medium containing MTT was removed, and a total of 100 µL dimethyl sulfoxide (DMSO) was added, followed by measuring the absorbance (490 nm) after 50 min inoculation.

### 4.7. Western Blot

Western blot was described previously [6]. Briefly, cells were lysed using Laemmli buffer containing 10% dithiothreitol (DTT, Thermo Fisher Scientific, Carlsbad, USA), and then denatured by heating at 95 °C for 5 min. Proteins were detected with antibodies against eIF4A (Cell Signaling Technology, Danvers, USA), eIF4E (Cell Signaling Technology), eIF4G (Cell Signaling Technology), PDCD4 (Cell Signaling Technology), and SA11 rotavirus VP4 (provided by professor Harry Greenberg, Stanford University School of Medicine, USA). The bound antibodies were visualized with Odyssey (LI-COR Biosciences, USA). The β-actin (Santa Cruz) was used as a loading control.

### 4.8. Statistics

All numerical results are reported as Mean ± Standard Error of Mean (SEM). The statistical significance of differences between means was assessed with the Mann–Whitney test using GraphPad Prism 5 (stationary-pc/mac version; GraphPad Software Inc., La Jolla, CA, USA). The statistical significance was defined as *P* ≤ 0.05.

## Figures and Tables

**Figure 1 ijms-20-01580-f001:**
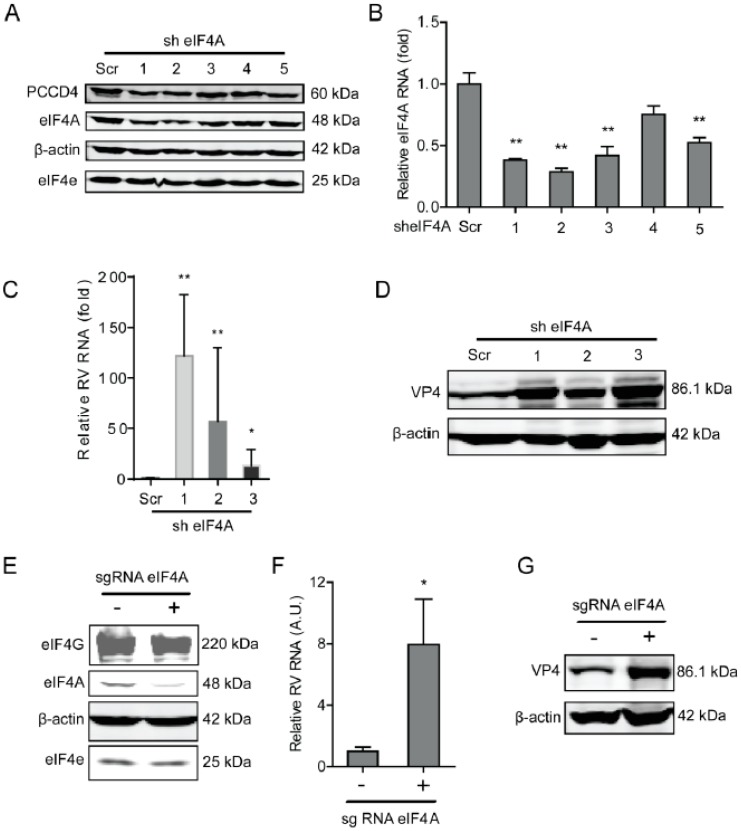
Knockdown of eIF4A supports rotavirus infection. (**A**) Western blot assay detected eIF4A in Caco2 cells transduced with lentiviral RNAi vectors against eIF4A. (**B**) Successful knockdown by shRNA vectors measured by RT-qPCR (*n* = 6, ** *P* < 0.01, Mann–Whitney test). (**C**) Silence of eIF4A increased rotavirus genomic RNA level (*n* = 6, * *P* < 0.05, ** *P* < 0.01, Mann–Whitney test). (**D**) Silence of eIF4A increased viral VP4 protein level in SA11 rotavirus infected Caco2 cells. (**E**) Western blot detected the eIF4A protein level in Caco2 cells transduced with CRIPSRs against eIF4A, suggesting a successful knockout. (**F**) Silence of eIF4A by CRISPR/Cas9 increased the rotavirus genomic RNA level (*n* = 6, * *P* < 0.05, Mann–Whitney test). (**G)** Silence of eIF4A by CRISPR/Cas9 increased the viral VP4 protein level in SA11 rotavirus infected Caco2 cells.

**Figure 2 ijms-20-01580-f002:**
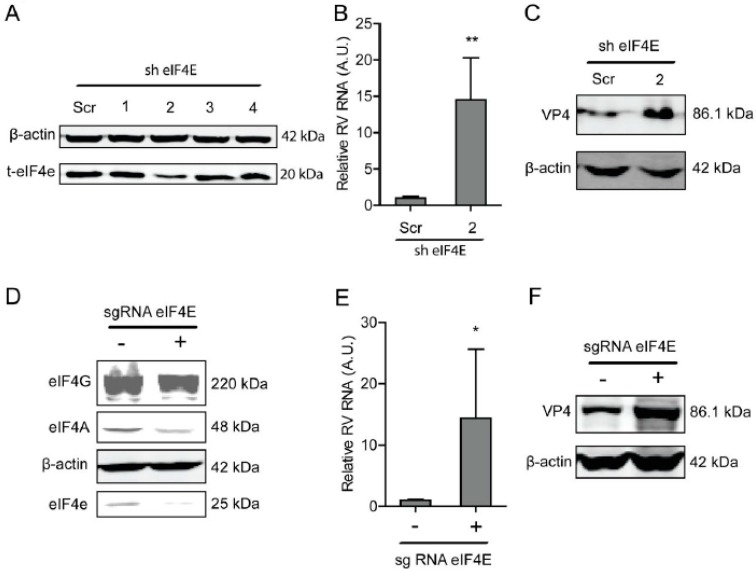
The eIF4E suppresses rotavirus infection. (**A**) Western blot assay measured the eIF4E protein in Caco2 cells transduced lentiviral RNAi vectors against eIF4E, indicating that one out of three shRNA vectors exerted a potent knockdown. (**B**) The knockdown of eIF4E significantly promoted rotavirus genomic RNA (*n* = 6, ** *P* < 0.01, Mann–Whitney test). (**C**) Western blot assay confirmed that eIF4E knockdown increased rotavirus protein level. (**D**) The eIF4E was knocked out by CRISPR/Cas9 assay. (**E**) Silence of eIF4E by CRISPR/Cas9 increased the rotavirus RNA level (*n* = 6, * *P* < 0.05, Mann–Whitney test). (**F**) Silence of eIF4E by CRISPR/Cas9 increased the rotavirus VP4 protein level.

**Figure 3 ijms-20-01580-f003:**
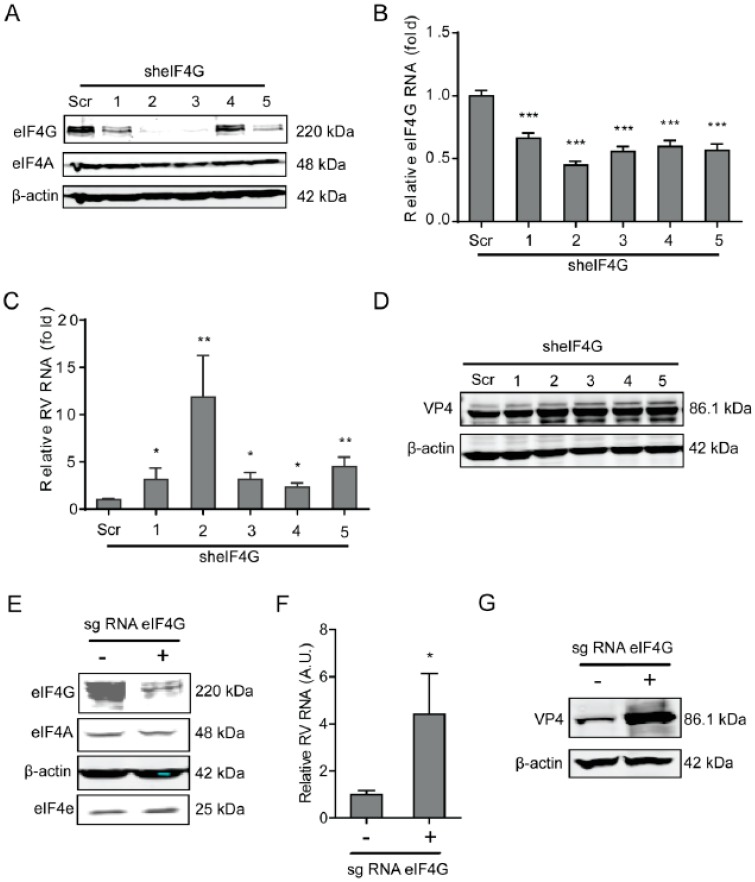
Silence of eIF4G promotes rotavirus infection. (**A**) Western blot assay detected eIF4G in Caco2 cells transduced with lentiviral RNAi vectors against eIF4G. (**B**) Successful knockdown by shRNA vectors detected by RT-qPCR assay (*n* = 6, *** *P* < 0.001, Mann–Whitney test). (**C**) Silence of eIF4G showed an increased rotavirus genomic RNA level (*n* = 6, * *P* < 0.05, ** *P* < 0.01, Mann–Whitney test). (**D**) Silence of eIF4G promoted the viral VP4 protein level in SA11 rotavirus infected Caco2 cells. (**E**) Western blot assay detected eIF4G in Caco2 cells transduced with CRIPSRs against eIF4G. (**F**) Silence of eIF4A by CRISPR/Cas9 increased the rotavirus genomic RNA level (*n* = 6, * *P* < 0.05, Mann–Whitney test). (**G**) Silence of eIF4G by CRISPR/Cas9 increased the viral VP4 protein level in SA11 rotavirus infected Caco2 cells.

**Figure 4 ijms-20-01580-f004:**
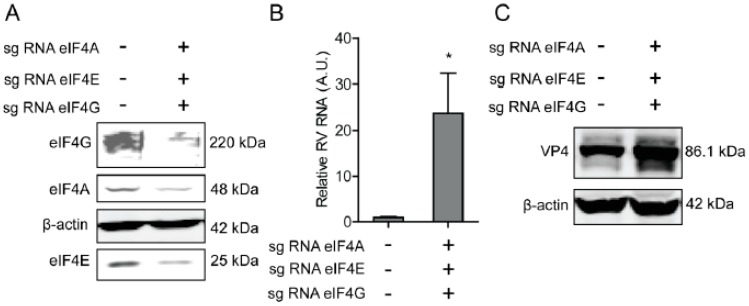
Silence of eIF4A/E/G promotes rotavirus infection. (**A**) Western blot assay detected eIF4 A/E/G in Caco2 cells transduced CRIPRs against eIF4A, eIF4E, and eIF4G. (**B**) Silence of eIF4A/E/G promoted the rotavirus genomic RNA level detected by qPCR assay (*n* = 6, * *P* < 0.05, Mann–Whitney test). (**C**) Silence of eIF4A/E/G showed increased rotavirus protein VP4.

**Figure 5 ijms-20-01580-f005:**
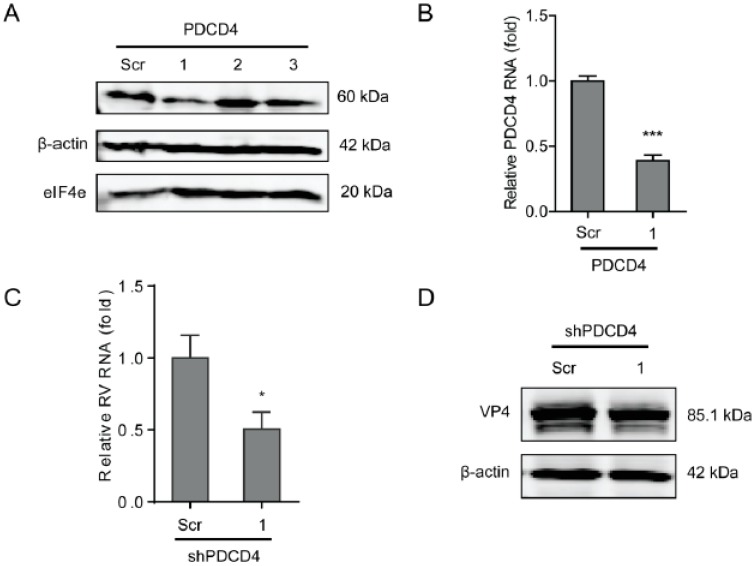
Programmed cell death protein 4 (PDCD4) sustains rotavirus infection. (**A**) Western blot assay detected PDCD4 in Caco2 cells transduced lentiviral RNAi vectors against PDCD4. (**B**) One out of three shRNA vectors showed successful knockdown detected by RT-qPCR (*n* = 6, *** *P* < 0.001, Mann–Whitney test). (**C**) Silence of PDCD4 showed inhibition of rotavirus genomic RNA levels (*n* = 8, * *P* < 0.05, Mann–Whitney test). (**D**) Silence of PDCD4 inhibited viral VP4 protein level in SA11 rotavirus infected Caco2 cells.

**Figure 6 ijms-20-01580-f006:**
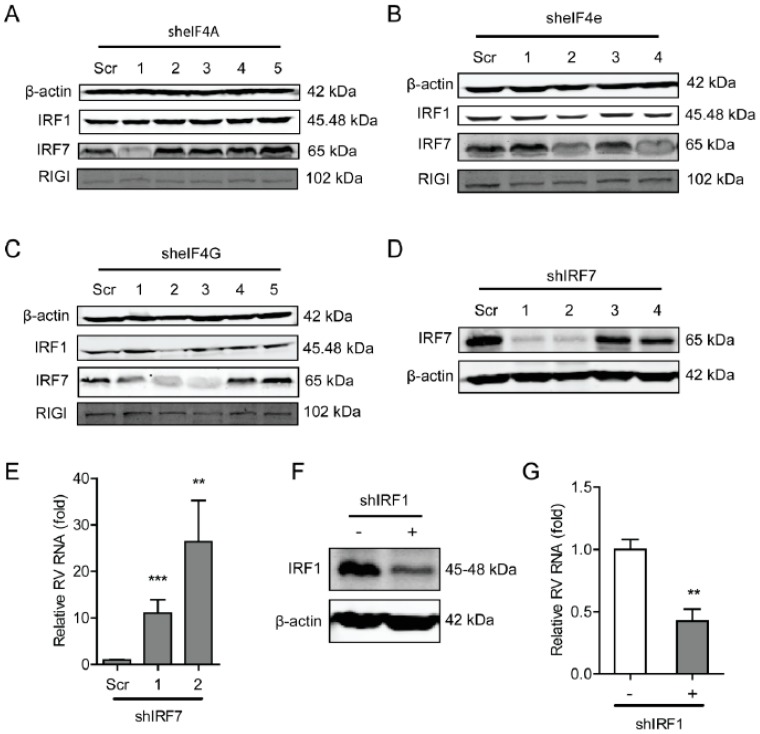
The eIF4F complex is required for IRF7 that exerts anti-rotavirus effect. (**A**) Western blot assay detected the protein level of antiviral factors including IRF1, IRF7, and RIG-I in eIF4A knockdown Caco2 cells. (**B**) Western blot assay detected the protein level of antiviral factors including IRF1, IRF7, and RIG-I in eIF4E knockdown Caco2 cells. (**C**) Western blot assay detected the protein level of antiviral factors including IRF1, IRF7, and RIG-I in eIF4G knockdown Caco2 cells. (**D**) Western blot assay detected the protein level of IRF7 in eIF4G knockout Caco2 cells, and two out of four shRNA vectors showed successful knockdown of IRF7 detected by western blot. (**E**) Silence of IRF7 increased rotavirus genomic RNA level detected by qPCR assay (*n* = 6, ** *P* < 0.01, *** *P* < 0.001, Mann–Whitney test). (**F**) Western blot assay detected the protein level of IRF7 in eIF4G knockout Caco2 cells, indicating the successful knockdown of IRF1 detected by western blot. (**G**) Silence of IRF1 increased the rotavirus genomic RNA level detected by qPCR assay (*n* = 6, ** *P* < 0.01, Mann–Whitney test).

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
