# Peer review of "The Eukaryotic Translation Initiation Factor 4F Complex Restricts Rotavirus Infection via Regulating the Expression of IRF1 and IRF7"

_ijms, 2019, doi:10.3390/ijms20071580_

Reviewer 1 Report

The author improved the manuscript by supplementing potential mechanisms. However, knock down data is confusing in several aspects. In figure 1B, the IF4A mRNA level decrease significantly, while I could not see too much change in protein level. A protein statistic level makes more sense than mRNA level here. Moreover, no mRNA level were shown during knocking down of IF4E. In figure 3A and 3B, the mRNA level decreased half while the protein level decreased multiple folds. There seems to be discrepancy among. Considering each protein was knock down into different level and most of the shRNA is not working well, the result of figure 6 is not convincing.

The discussion part could solve these concerns and related questions arose from the manuscript, instead of introducing more background.

The written language need to be improved, especially for line 25 to line 27 in the abstract.

Author Response

1. we thank the reviewer for his/her comments and suggestions, for mismatch of mRNA level and protein level, to our best knowledge, many groups have observed this kind of discrepancy, for instance, in the review article written by Maier et al. (FEBS Lett. 2009 Dec 17;583(24):3966-73. PMID: 19850042), it was mentioned that the correlation between mRNA and protein abundances in the cell has been reported to be notoriously poor, which depends on various biological and technical factors. Nevertheless, the tendency of changing of mRNA and protein expression in our study is same, indicating that reliability of our results.

 2. Regarding the mechanism underlying the effects of eIF4F complex on rotavirus infection, we tested effects of eIF4F on several antiviral proteins, and found that the IRF1 and IRF7 might be most relevant. Of note, we demonstrated that IRF1 and IRF7 per se obviously inhibit rotavirus infection. However, the mechanism may be quite complicated, and more efforts should be investigated. Thus, we will invest more efforts to unravel this difficulty. 

Reviewer 2 Report

The authors significantly improved the manuscript edition as well as added deeply scientific explanation their results especially in disscusion section.

Article are adjust to the journal rules - but material and methods section should be definatively after disscusion , not after introduction. The results are more structured, moreover, the data presention is better organised- more clear.

Therefore, in my opinion the article can be accepted to publication in Int. J. Mol. Sci. after minor text editing.

Author Response

We thank the reviewer for his/her endorsement on our manuscript, and materials and methods have been revised accordingly.

This manuscript is a resubmission of an earlier submission. The following is a list of the peer review reports and author responses from that submission.

Round  1

Reviewer 1 Report

Dear authors, in this manuscript you found knock down/out either eIF4A, eIF4G or eIF4E could help to increase the expression of VP6 mRNA and VP4 protein expression. First, this study has no physiological relevance. Based on that, there is no mechanism study. Secondly, the conclusion could not drew from the data. The data does not support that any of the factors directly suppress rotavirus gene expression. Since eIF4 complex is important for so many genes expression, any of the downstream event could result in the observations. Moreover, since all the knock down/knock out in this manuscript improves the rotavirus gene expression, there is not a control gene that by knocking down will decrease its expression. The statistics and calculation seems intrigue. If we compare Fig. 1A and 1E, eIF1A/β-actin changed from 0.92 to 0.12 comparing scramble shRNA treatment and control sfRNA knock down. This problem exist in Fig. B and C. In Fig.3A, the expression of eIF4G is really minor, while the eIF4G/β-actin reading is still comparable to other figures. These data need to be explained in more detail. Overall, I could not find significant novelty from this manuscript and the design could be improved. 

Reviewer 2 Report

Dear Authors,

I have a great honor to review manuscript ijms-382844, entitled: “The eukaryotic translation initiation factor 4F 3 complex restricts rotavirus replication” which is consider for publication in International Journal of Molecular Sciences, section Molecular Microbiology. I carefully read and analyzed whole manuscript and in my opinion it presents some interesting results but has also many serious flaws especially in sloppy preparation of main text of article. Moreover, with heavy heart I must say that authors did not pay almost any attention to journal rules. I would like to prepared precise list of critical comments to the article:

Line 16: Section “Abbreviations:” must be absolutely deleted because such list is not acceptable by journal rules in this place. This rules are exact and precise from many years. All abbreviations should be explained in main text in sections where they first appeared. In some places in text abbreviations are correctly explained but in other they are not explained at all. All abbreviations should be careful analyzed and corrected along whole article.

Abstract

Line 26: As I mentioned above all abbreviations should be exactly explained in abstract section. Abstract is too long (please see journal rules) and should be more concise. Authors should rewrite or shorten abstract.

Line 36:  “…viral protein”- authors should precise from abstract which exactly viral protein they have in their mind. There is commonly known fact that abstract must briefly presenting the article results.

Line 38:  “…. host factor in defending rotavirus replication..”- term of “defending ” is rather incorrect and little misleading. Because authors did not present any model of or proposition of model of such defense. Better term will be “host factor of great importance in rotavirus replication”.

Line 39 : ” …Rotavirus-host interactions” I suggest to delete this term because authors mainly did not show rotavirus host interaction in the cell (host cell localization).

Introduction

General statement to this section: Authors should change order of presentation in this section. Firstly should be presented virus and in this part authors should add more information about VP4 protein which the authors localized by using the Western Blot technique, moreover, the amount of information about VP4 is absolutely unacceptable in current form of the manuscript. After presentation of virus authors should present eIFs and their roles.

Line 45-47: Term “m7GpppX” must be explained fully in this line not partially. Other explanation to this term are illogically dispersed in introduction. This fact make this section difficult to read.

Line 65: I suggest do not use term “progeny virus” more acceptable is “new synthesized virus. Progeny is mainly connected with generative reproduction and viruses simply did not have it.

Results

General statement to this section: All results are prepared only for Simian rotavirus SA11, which as I assume is only one strain/isolate (??) of rotavirus. Is it a model rotavirus (maybe??) which leads to the conclusion about the whole rotaviruses group ?? Rotavirus is very variability in populations and this fact is crucial element which indicates its danger for humankind. From results, I think will be better to compare other strain of the virus to asses all of this dependence between rotavirus and different eIF. Moreover, on some Figures (2, 3, 4,5) I observed many pixels, which suggest small resolution of photos and it should be definitely corrected.

Line 104:  term shRNA (??) should be explained in text of results. As reviewer I know this term, but other readers may have with this a problem

Line 107-108: Western blot assay 108 indicated that shRNA vector No 1, 2 and 3 promoted rotavirus VP4 protein synthesis…”- How authors measured by using of Western Blot technique promotion of VP4 synthesis. VP4 protein level was higher??? If so, which method was used to measure amount of viral protein in cells??

 Discussion

General statement to this section: Discussion in general is very problematic and difficult to read, because is too short in comparison to the result in normal and supplementary materials. What’s worse, again authors did not pay any attention to journal publication rules. Order of citation from last part of introduction to discussion is incorrect. ALL citations in International Journal of Molecular Sciences are ordered in sequence from 1,2,3 etc. Introduction section ends on [23] - but discussion from unknown reason starts from [13,29]. In beginning of discussion section authors did not cited [24,25,26,27,28]. From more strange reasons references [24-28] are in last section with methods. Order of references in whole manuscript must be checked and corrected. If introduction ends on 23 then discussion starts from 24 of course and so on.

Material and Methods

Line 277: If authors made citation [27,28] they cannot add name of http address of website. If 27 and 28 is articles about method then website should have her own number of citation for example [26] and then citations should be presented in this way: “….using freely available online tools [26,27,28]”

Line 308-310: In statistic section authors should explain abbreviation SEM because this abbreviation in this form of writing has many meanings. Authors must add information about versions of GraphPad, was this online version of software ? or it was stationary-pc/mac version (journal rules)?  If it was online then authors should add citation of website as I mentioned in Line 277 comment.

 References

In this section surprising is that the any of references in this list is correct. Once again I must indicate to journal rules. Citation ways in International Journal of Molecular Sciences are exactly and clearly presented on journal website. And also information about DOI number.